# Effect of Palm Kernel Meal and Malic Acid on Rumen Characteristics of Growing Naemi Lambs Fed Total Mixed Ration

**DOI:** 10.3390/ani9070408

**Published:** 2019-07-01

**Authors:** Mutassim M. Abdelrahman, Ibrahim Alhidary, Hani H. Albaadani, Mohsen Alobre, Rifat Ullah Khan, Riyadh S. Aljumaah

**Affiliations:** 1Department of Animal Production, College of Food and Agriculture Sciences, King Saud University, P.O. Box 2460, Riyadh 11451, Saudi Arabia; 2Department of Animal Health, Faculty of Animal Husbandry & Veterinary Sciences, The University of Agriculture Peshawar, Peshawar 25000, Pakistan

**Keywords:** complete feed, growing lambs, palm kernel cake, organic acids, rumen characteristics

## Abstract

**Simple Summary:**

To increase the feed efficiency, animals are usually fed with total mixed ration (TMR), however, its consumption is also associated with lactic acidosis. To minimize the incidence of lactic acidosis, different feeding systems are used. In this study, we have used malic acid to reduce the clinical signs of lactic acidosis in a total of 32 growing lambs. Lambs were either fed a control (barley and alfalfa ha) diet (control) or TMR (T1), TMR + 20% palm kernel meal (T2), TMR + palm kernel meal 20% + 4 mL/day malic acids (T3). The results showed that propionic acid in the rumen fluid increased in T1 and T3, while lactic acid increased in T2. Rumen and reticulum discoloration were improved in T3 while histomorphological features were higher in T3 and T2. We concluded that the addition of malic acid supplementation to lambs fed PKC caused a significant improvement in the rumen pH and decreased lactic acid concentration in growing Naemi lambs.

**Abstract:**

This study was conducted to investigate the effect of malic acid and 20% palm kernel meal (PKM) on ruminal characteristics. A total of 32 growing lambs were randomly distributed into control (barley and alfalfa ha), total mixed ration (T1), TMR + 20% PKM (T2), TMR + PKC 20% + 4 mL/day malic acids (T3). Lambs were fed these diets ad libitum for 84 days. The results showed that propionic acid in the rumen fluid increased significantly (*p* < 0.05) in T1 and T3. Lactic acid concentration of rumen fluid increased significantly (*p* < 0.05) in T2 while the pH increased significantly. The coloration of rumen and reticulum was improved in T3. In addition, most of the histomorphological features were higher in T3 and T2. We concluded that the addition of malic acid supplementation to lambs fed PKC caused a significant improvement in the rumen pH and decreased lactic acid concentration in growing Naemi lambs.

## 1. Introduction

Feeding total mixed rations (TMR) with highly fermentable carbohydrates may increase the risk of subacute rumen acidosis by depressing the urinal pH and, consequently, affect ruminant animal health and productivity. The fiber portion of the TMR is believed to be effective in stimulating rumination, chewing activities, and salivary production, which acts as a buffer [1,2,3]. Palm kernel meal (PKC) can be economically used compared to other TMR ingredients [4] as a good source of neutral detergent fibers and crude protein [5]. Alhidary et al. [6] reported that several factors such as pelleting, iron concentration, particle size, and types of buffer in the feed affect the color of rumen and reticulum. Alhidary et al. [6] and Abdelrahman et al. [7] reported a significant change in rumen characteristics including a very dark rumen tissues coloring and general performance in growing lambs fed TMR consisted of PKM compared with the traditional feeding system (barley and alfalfa hay). 

Organic acids have been documented to promote higher rumen pH by reducing lactic acid production and stimulating ruminal lactate utilizing bacteria in ruminant fed high concentrate diet [8,9,10]. Malic acid increases total volatile fatty acids, propionate to acetate ratio, and reduces methane production by sinking hydrogen in the rumen [11]. In addition, malic acid may affect rumen digestion process by shifting protein breakdown and other dietary ingredients and eliminates the ammonia odor. The purpose of this study was to find the effect of the supplementation of malic acid and 20% palm kernel meal on rumen characteristics of Naeemi lambs.

## 2. Materials and Methods

The study was approved by the committee on ethics and animal welfare, King Saud University, Riadh, Saudi Arabia.

### 2.1. Animals Selection and Experimental Design

A total of 32 apparently healthy growing Naemi male lambs, about three months old, were randomly selected. Lambs were housed in individual pens and vaccinated against enterotoxemia and endoparasites. After 15 days of adaptation, the lambs were randomly divided into four dietary treatments. The dietary treatments included a control (barley and alfalfa hay), TMR (T1), TMR with PKM 20% (T2), TMR + PKM 20% + 4 mL/ day malic acid (T3). Lambs were fed these diets ad libitum for 98 days. The feed composition of each group is given in Table 1. Malic acid was purchased from Sigma-Aldrich Chemical Company (St. Louis, MO, USA). Feed and PKM samples were collected weekly for nutritional analysis [6,12].

### 2.2. Rumen pH

Rumen fluid samples were collected on a weekly basis from three lambs/group regularly at zero, two, four and eight hours post feeding using a special esophageal silicon tube connected to vacuum pump from the ventral rumen sac. The pH of the fluid was measured immediately with a portable pH-meter. Samples were filtered through four layers of cheesecloth to discard the solid unfermented particles in order to obtain 50 ml of the ruminal fluids. Then 10 mL subsamples were preserved by the addition of 0.2 ml of 50% sulfuric acid and frozen at −20 C° until subsequent analysis.

### 2.3. Colour Intensity of Rumen and Reticulum

At the end of the experiment, three lambs were randomly selected from each treatment group and slaughtered after effective stunning. After slaughtering, rumen and reticulum samples were collected for the coloring evaluation process. The color intensity was measured for the rumen and reticulum by using a Chroma meter (Konica Minolta, CR-400-, Tokyo, Japan) and CIELAB Color System (1976) for the color values (L* for lightness, a* for redness and b* for yellowness).

### 2.4. Histology of Rumen

Histology of rumen was performed following the method of Alhidary et al. [6]. Rumen and reticulum tissues samples were collected from the ventral sacs. All the samples were processed and stained with hematoxylin and eosin (H&E) dye. The slides of the tissues were studies under Inverted Olympus Microscope attached with a PC-based image analysis system (Olympus DP72 microscope digital camera, Olympus NV, Aartselaar, Belgium) with software analysis (cellSens digital imaging software for research application, Olympus).

### 2.5. Volatile fatty Acids Determination

Rumen fluid samples were collected regularly at 0, 2, 4 and 8 h after feeding to be analyzed for acetate, propionate, butyrate and lactic acid concentration by using an Agilent series gas chromatograph with an AFFAP special capillary column (Agilent Technologies Inc., Wilmington, DE, USA) based on methodology from Supelco Inc (Bellefonte, PA, USA).

### 2.6. Statistical Analysis

Data were subjected to analysis of variance (ANOVA) using the general linear model (GLM) procedure of Statistical Analysis System Institute, Inc. (Cary, NC, USA) [13]. Means of each treatment were compared by using protected least significant differences (LSD) at *p* < 0.05.

## 3. Results

Effect of treatments and times VFAs in rumen fluid of the growing Naemi lambs is shown in Table 2. The results of this study show a significant increase in propionic concentration in the rumen fluid of lambs from the control, T1 and T3 groups compared with the lambs from T2. This means that organic acid led to decreased propionic acid concentration and negatively affected the rumen digestion and absorption process. Moreover, there is no significant difference between all groups in terms of acetic and butyric acid levels in the rumen fluid, even though the acetic acid level was numerically higher in lambs supplemented with organic acid compared with other groups. In addition, a significant increase was found in the concentration of propionic and butyric acid after 8 h. 

The effect of treatments and times on lactic acid and pH of rumen fluid in the lambs is shown in Table 3. The lactic acid concentration was significantly (*p* < 0.05) higher in T2 compared to the rest of the treatments. In addition, the lactic acid value decreased significantly (*p* < 0.05) after 8 h. Significantly (*p* < 0.05) lower pH was found in T2 compared to the rest of the treatments. Similarly, low pH was also found after 2 h and then steadily increased. Malic acid supplementation to lambs fed PKC caused a significant improvement in the rumen pH and did not differ compared with the values for the T1 group. Varied effects between treatments were reported for the pH values, but all the pH values were within the normal healthy recommended levels.

The results of the color changes in rumen and reticulum in all dietary groups of growing lambs are presented in Table 4. The rumen tissue of the growing lambs fed on TMR with 20% PKM showed black color compared with all other dietary groups. On the other hand, the drenching of organic acid to growing lambs fed PKM significantly improved the rumen color (lightness values) compared to lambs from T2. There was no significant difference in reticulum color in T1, T2, and T3, but it was significantly darker when compared to lambs from the control group.

Histomorphometric measurements of the rumen of the growing lambs were affected (*p* < 0.001) by dietary groups except for stratum corneum, epithelium width, and submucosa (Table 5). The results indicate that papillae length and surface area improved in T3 compared to the other treatment groups. On the other hand, papillae width and density were significantly lower in dietary groups T2 and T3 as compared with T1 and control. The total surface of papillae was significantly lower in dietary groups T2 and T3 due to the reduced number of papillae (density) as compared with other groups. In addition, lamina propria was significantly lower in all the treatment groups.

The histomorphology changes of the rumen from different dietary groups are presented in Figure 1. Lambs fed on the control diet showed that the height of papillae is not regular where most of the short papillae and lamina propria layers are wider with a mild, thick layer of keratinization. Dietary group of TMR without PKM (T1) showed that papillae height and width have a regular papillae length and formation of new papillae structures.

## 4. Discussion

In the current study, no significant change was found in the concentration of acetic and butyric acid, however, propionic acid decreased significantly in the group fed with TMR and PKM. Similarly, the pH was decreased significantly in the same group. It is generally reported in many research findings that ruminal pH increased with malic acid supplementation by manipulating the rumen microbial fermentation process [14,15,16]. These findings support the idea that PKM causes a negative effect on the rumen digestion process affecting the pH and VFAs balance. Many researchers reported significant improvements in lactic acid utilizing bacteria, increase pH, total volatile fatty acids production and propionic levels in the rumen as a result of malic acid supplementation [17,18]. These findings completely agree with our results of this study. In the current study, the concentration of lactic acid increased in T2, however, rumen pH decreased in the same group. The probable reason is that rumen pH is the result of the overall concentration of individual VFAs, which either did not differ among the treatments or were significantly low in the T2 in the present study. Secondly, it is obvious that the higher the production of fatty acids, the lower the pH of the rumen is.

In the current study, rumen and reticulum color was improved by the addition of malic acid in TMR. Many factors cause a change in the rumen and reticulum color, such as pelleting, fiber particle size content, dietary iron levels, and the type of buffering system [6]. Feeding fattening lambs a high fermentable TMR may increase the risk of acidosis by dropping the rumen pH levels as a result of increased lactic acid production in the rumen fluid, resulting in a darker color of the rumen epithelium tissues [6,7,19]. Moreover, the possible dark color may be caused by consuming high iron levels by fattening lambs fed TMR forming a black stain in the rumen epithelium tissues [6,20].

In the current study, the epithelial height and surface area were increased in T3 while other parameters such as width, surface area, density, and the total surface of papillae were higher in T1. In general, the papillae size, surface and, other features are mainly affected by levels of volatile fatty acids, especially propionic and butyric, and pH levels which are mainly affected by organic acid supplementation in the present study and cause a significant positive effect on the papillae characteristics. Shen et al. [21], Suarez et al. [22] and Alvarez-Rodriguez et al. [23] reported an increased surface area of papillae with high production of volatile fatty acids. Moreover, propionic acid significantly increases in this study in the group supplemented with organic acid, which might have positively affected the rumen papillae size, surface and other features which are consistent with findings of Blanco et al. [24].

## 5. Conclusions

The results of the present study show that the inclusion of malic acid in total mixed ration improved the dimensions of the epithelium and reduced the discoloration of the rumen and reticulum.

## Figures and Tables

**Figure 1 animals-09-00408-f001:**
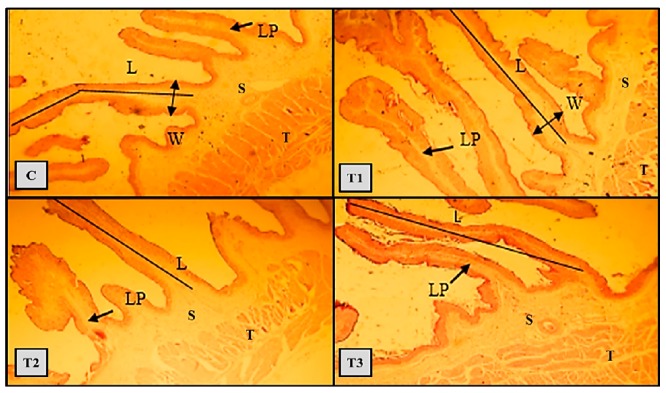
Histomorphology of lambs’ rumen during slaughter (the end of experiment) H&E, (×100). L = papilla height, W = width, LP = lamina propria, S = submucosa and T = tunica muscular. Treatments: C = barley and alfalfa hay were showing short and irregular papillae, lamina propria layer is wide with a mild thick layer of keratinized whereas, layers S and T were normal. T1 = TMR1 without PKM showed that L and W were within the norm, we could see regular papillae and formation of new structures. T2 = TMR2 with 20% PKC showed histological changes in the papillae structure, whereas T3 = T MR2 with 4 mL/ day drench with organic acids showed a clear improvement in the structure of the papillae.

**Table 1 animals-09-00408-t001:** Ingredients and chemical composition of the experimental diets.

Ingredients, %	Treatments ^4^						
	AH ^1^	BG ^2^	PKM ^3^	Control	T1	T2	T3
Alfalfa Hay				25.22	-	-	-
Barley				74.78	27.0	17.0	17.0
Feed Wheat				-	29.95	29.92	29.92
Wheat Bran				-	5	5	5
Sunflower Meal				-	17.35	10.05	10.05
Soya Hulls				-	13.55	11.03	11.03
Palm Kernel Meal				-	0	20	20
Salt				-	0.54	0.47	0.47
Limestone				-	2.51	2.58	2.58
Molasses				-	3	3	3
Acid buffer^5^				-	0.95	0.80	0.80
Commercial Premix^6^				-	0.15	0.15	0.15
Organic acid				-	-	-	4 mL/day
Dry matter	93.44	90.49	94.56	91.23	90.8	91.40	91.40
Crude protein	15.69	12.79	15.33	13.53	13.24	13.79	13.79
Crude fat	26.01	5.70	16.58	10.82	12.72	11.98	11.98
Ether extract			11.27		1.40	2.61	2.61
Ash	11.20	2.93	4.11	5.02	10.30	9.09	9.09
ME, Mcal/kg^7^	2.03	3.11	2.66	2.83	2.80	2.79	2.79
Macro-minerals, %							
Calcium	1.41	0.05	0.31	0.39	1.90	1.7	1.7
Phosphorous	0.24	0.38	0.50	0.35	0.39	0.42	0.42
Magnesium	0.31	0.15	0.23	0.19	0.38	0.28	0.28
Potassium	1.71	0.47	0.44	0.78	0.78	0.75	0.75
Micro-minerals, mg/kg							
Iron	134	85	801	97	193	336	336
Copper	14	9	21.6	10.3	12.4	26.8	26.8
Zinc	23	19	34.5	20.0	39.3	269	269
Manganese	28	18	259	20.5	133	99.2	99.2
Selenium	-	0.22	-	0.16	0.48	0.25	0.25

^1^ AH = alfalfa hay. ^2^ BG = barley grain. ^3^ PKM = palm kernel meal. ^4^ Control = traditional feeding protocol (loose whole barley grain + alfalfa hay), T1 (0% PKM), T2 = 20% PKM and T3 = 20% PKM plus 4 ml organic acid/lamb/day drenching. ^5^ Neutral buffer derived from seaweed (celtic Sea Company, Ireland). ^6^ Contained per kg, 10000 IU vitamin A, 1000 IU vitamin D, 20 IU vitamin E, 300 mg Mg, 24 mg Cu, 0.6 mg Co, 1.2 mg I, 60 mg Mn, 0.3 mg Se, 60 mg Zn. Vitamin A 3335000 IU/kg, Vitamin D 335000 IU/kg, Vitamin E 16670 IU/kg, Cobalt 200 mg/kg, Copper 1600 mg/kg, Iodine 500 mg/kg, Iron 0.0 mg/kg, Magnesium 100000 mg/kg, Manganese 10000 mg/kg, Selenium 100 mg/kg, Zinc 33340 mg/kg. ^7^ Based on tabulated values.

**Table 2 animals-09-00408-t002:** Effect of treatments and time volatile fatty acids (VFA) in rumen fluid of the growing Naemi lambs.

Variables	VFA (g)		
	Acetic	Propionic	Butyric
Treatment ^1^			
Control	1515.9	1690.2 ^ab^	551.7
T1	1664.6	1923.2 ^a^	545.5
T2	1774.4	1414.4 ^b^	676.2
T3	1941.3	1989.2 ^a^	517.7
SEM ^2^	154.641	127.783	47.489
*p*-value	0.2767	0.0136	0.1066
Time (h)			
0	1424.2	1412.5 ^b^	435.4 ^b^
2	1981.3	1724.3 ^ab^	596.4 ^a^
4	1661.9	1948.6 ^a^	602.6 ^a^
8	1828.7	1931.6 ^a^	656.7 ^a^
SEM^2^	154.640	47.489	127.783
*p*-value	0.0878	0.0191	0.0151
Treatment × Time			
*p*-value	0.7990	0.0188	0.7232

^1^ Treatments: Control = barley and alfalfa hay, T1: TMR without PKM, T2: TMR with 20% PKM and T3: TMR with 4 mL/day malic acid. ^2^ SEM: standard error of the mean. ^a,b^ Means values within columns with different superscripts are significantly different (*p* < 0.05).

**Table 3 animals-09-00408-t003:** Effect of treatments and times on lactic acid and pH in rumen fluid of the growing Naemi lambs at 42 days.

Variables	Parameters	
	Lactic Acid	pH
Treatment ^1^		
Control	1.94 ^b^	7.23 ^a^
T1	1.82 ^b^	6.55 ^b^
T2	3.83 ^a^	6.34 ^c^
T3	1.86 ^b^	6.42 ^b^
SEM ^2^	0.2924	0.1045
*p*-value	<0.0001	<0.0001
Times (h)		
0	2.63 ^a^	7.07 ^a^
2	2.62 ^a^	6.18 ^c^
4	2.62 ^a^	6.72 ^b^
8	1.59 ^b^	6.56 ^b^
SEM^2^	0.2920	0.1040
*p*-value	0.0405	<0.0001
Treatment × Time		
*p*-value	0.2150	0.2460

^1^ Treatments: Control = barley and alfalfa hay, T1: TMR1 without PKM, T2: TMR2 with 20% PKM and T3: TMR2 with 4 mL/day malic acid. ^2^ SEM: standard error of the mean. ^a,b^ Means values within columns with different superscripts are significantly different (*p* < 0.05).

**Table 4 animals-09-00408-t004:** Effect of treatments on the color of rumen and reticulum of the growing Naemi lambs (the end of the experiment).

Parameter ^3^		Treatments ^2^				SEM ^1^	*p*-Value
		Control	T1	T2	T3		
Rumen	L*	53.81 ^a^	33.38 ^b^	32.19 ^c^	35.24 ^b^	0.807	<0.0001
	a*	4.94 ^a^	2.20 ^b^	2.63 ^b^	4.63 ^a^	0.265	0.0002
	b*	15.06 ^a^	6.63 ^bc^	6.34 ^c^	7.89 ^b^	0.449	<0.0001
Reticulum	L*	62.09 ^a^	43.21 ^b^	42.90 ^b^	38.18 ^b^	2.512	0.0007
	a*	4.60	2.91	3.47	3.39	0.480	0.1622
	b*	15.75 ^a^	8.00 ^b^	8.39 ^b^	7.61 ^b^	1.085	0.0020

^1^ SEM: standard error of the mean. ^2^ Treatments: Control = barley and alfalfa hay, T1: TMR1 without PKM, T2: TMR2 with 20% PKM and T3: TMR2 with 4 mL/day malic acid. ^3^ L* = lightness; a* = redness; b* = yellowness. ^a,b,c^ Means values within a row with different superscripts are significantly different (*p* < 0.05).

**Table 5 animals-09-00408-t005:** Effect of treatments on histomorphometric of the rumen of the growing Naemi lambs (the end of the experiment).

Parameter ^3^	Treatments ^2^				SEM ^1^	*p*-Value
	Control	T1	T2	T3		
L (mm)	3.73 ^bc^	4.67 ^ab^	2.75 ^c^	5.49 ^a^	0.375	0.005
W (mm)	0.75 ^a^	0.67 ^ab^	0.49 ^c^	0.56 ^bc^	0.035	0.004
SA (mm^2^)	8.84 ^a^	10.00 ^a^	4.20 ^b^	9.89 ^a^	0.971	0.009
Density (N/cm^2^)	77.79 ^a^	70.33 ^ab^	64.66 ^bc^	56.33 ^c^	3.791	0.022
TSP (mm^2^/cm^2^)	436.67 ^a^	439.99 ^a^	172.35 ^c^	348.09 ^b^	20.742	<0.0001
SC (mm)	0.041	0.032	0.046	0.040	0.005	0.335
WE (mm)	0.205	0.190	0.141	0.196	0.0181	0.133
LP (mm)	0.374 ^a^	0.241 ^b^	0.162 ^b^	0.183 ^b^	0.031	0.006
Submucosa (mm)	0.761	0.814	0.648	0.608	0.069	0.208

^1^ SEM: Standard error of the mean. ^2^ Treatments: Control = barley and alfalfa hay, T1: TMR1 without PKM, T2: TMR2 with 20% PKM and T3: TMR2 with 4 mL/day malic acid. ^3^ Parameter: L = papilla height, W = width, SA = papilla surface area, TSP = the total surface of papillae, SC = stratum corneum, WE = the width of epithelium and LP = lamina propria. ^a,b,c^ Means values within a row with different superscripts are significantly different (*p* < 0.05).

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
