# Peer review of "Effect of Palm Kernel Meal and Malic Acid on Rumen Characteristics of Growing Naemi Lambs Fed Total Mixed Ration"

_animals, 2019, doi:10.3390/ani9070408_

Round 1
Reviewer 1 Report
All the revision is in the pdf file attached

Author Response
Authors
The Title should be improved, it must be more related to the study.
For example: Effect of palm kernel meal and malic acid over rumen characteristics of growing Naemi lambs fed total mixed ration.
Response: corrected
Simple summary;
Response: corrected
Line 14, add after (barley and alfalfa hay) diet, (control)
Response: added
Line 16, separate increased in / in increased in
Response: corrected
Abstract
Line 21, change rumen by ruminal
Response: corrected
Lines 28 to 29, conclude about rumen fermentation characteristics, for example “Malic acid supplementation to lambs fed PKC caused a significant improvement in the rumen pH and decreased lactic acid concentration”.
Response: Revised as suggested
Line 30, change Growing lambs by growing lambs
Response: corrected
Introduction
There is not enough evidence present in this section about how does Malic acid affect the rumen and reticulum coloration and histomorphological features in growing lambs that justify to study rumen characteristics.
Response: This is the novelty of this study that the effect of malic acid on the rumen coloration has not been studies before. Relevant review regarding the digestion and rumen pH though has been provided.
The study has an objective to investigate the effect of malic acid and 20% palm kernel meal on rumen characteristics, However, palm kernel meal is not considered in the hypothesis, please clarify.
Response: provided
Line 37 to 39, this paragraph “Alhidary et al. [6] reported that dietary minerals especially iron can affect the pigmentation of rumen that attached firmly to the keratinized layer of the epithelium tissues”, needs to connect with the following paragraph. Better explanation appears in discussion lines 148 to 152.
Response: improved
Materials and methods
Line 55, The study considered a total of 32 healthy growing Naemi male lambs, but do not explain which kind of methodology were considered to state that those animals were healthy.
Response: apparently healthy lambs were chosen.
Line 58, add after (barley and alfalfa hay) diet, (control)
Response: corrected
Line 59, add after TMR + 58 PKM 20% + 4ml/ day malic acid (T3)
Response: corrected
Line 59, 4ml/ day malic acid (Organic acid drenching), describe source, brand, concentration, etc.
Response: corrected
Response: MA was purchased from Sigma-Aldrich Chemical Company (St. Louis, MO).
Response: corrected
Line 62, state “rumen fluid samples were collected on weekly basis from three lambs/ group regularly at 0, 2, 62 4 and 8 hour posts feeding” and Line 83, state “rumen fluid samples were collected regularly before feeding and at 2, 4 and 8 hrs. after feeding”. Please clarify and explain “regularly”.
Response; At line 83, before feeding was used which means “0” hour.
There is not information for forage and supplement sampling and analyses.
Response: Provided
Results
Line 94, says C instead of control.
Response: corrected
Tables 2, 3 and 4 uses C for control and table 1, Control. Please be consistent.
Response: corrected
Table 2, report the unit that VFA are expressed.
Response: corrected
Line 102, state “lactic acid production was significantly (P<0.05) high in T2 compared to the rest of the treatments”, however, Line 84 says that rumen samples were “analyzed for acetate, propionate, butyrate and lactic acid concentration”. Clarify the figures presented in tables 2 and 3 are production or concentration.
Response: corrected
Discussion.
In consideration to the broad amount of results, I consider that discussion is insufficient. For example, there are many comparisons between T3 with T2 but there are not explanations among them. There are not reasons why T2 decrease pH and increase Lactic acid.
Response: relevant information were provided.
Reviewer 2 Report
The Title should be improved, it must be more related to the study.
For example: Effect of palm kernel meal and malic acid over rumen characteristics of growing Naemi lambs fed total mixed ration.
Simple summary;
Line 14, add after (barley and alfalfa hay) diet, (control)
Line 16, separate increased in / in increased in
Abstract
Line 21, change rumen by ruminal
Lines 28 to 29, conclude about rumen fermentation characteristics, for example “Malic acid supplementation to lambs fed PKC caused a significant improvement in the rumen pH and decreased lactic acid concentration”.
Line 30, change Growing lambs by growing lambs
Introduction
There is not enough evidence present in this section about how does Malic acid affect the rumen and reticulum coloration and histomorphological features in growing lambs that justify to study rumen characteristics.
The study has an objective to investigate the effect of malic acid and 20% palm kernel meal on rumen characteristics, However, palm kernel meal is not considered in the hypothesis, please clarify.
Line 37 to 39, this paragraph “Alhidary et al. [6] reported that dietary minerals especially iron can affect the pigmentation of rumen that attached firmly to the keratinized layer of the epithelium tissues”, needs to connect with the following paragraph. Better explanation appears in discussion lines 148 to 152.
Materials and methods
This section appears insufficient to replicate the study.
Line 55, The study considered a total of 32 healthy growing Naemi male lambs, but do not explain which kind of methodology were considered to state that those animals were healthy.
Line 58, add after (barley and alfalfa hay) diet, (control)
Line 59, add after TMR + 58 PKM 20% + 4ml/ day malic acid (T3)
Line 59, 4ml/ day malic acid (Organic acid drenching), describe source, brand, concentration, etc.
Line 62, state “rumen fluid samples were collected on weekly basis from three lambs/ group regularly at 0, 2, 62 4 and 8 hour posts feeding” and Line 83, state “rumen fluid samples were collected regularly before feeding and at 2, 4 and 8 hrs. after feeding”. Please clarify and explain “regularly”.
There is not information for forage and supplement sampling and analyses.
Results
Line 94, says C instead of control.
Tables 2, 3 and 4 uses C for control and table 1, Control. Please be consistent.
Table 2, report the unit that VFA are expressed.
Line 102, state “lactic acid production was significantly (P<0.05) high in T2 compared to the rest of the treatments”, however, Line 84 says that rumen samples were “analyzed for acetate, propionate, butyrate and lactic acid concentration”. Clarify the figures presented in tables 2 and 3 are production or concentration.
Discussion.
In consideration to the broad amount of results, I consider that discussion is insufficient. For example, there are many comparisons between T3 with T2 but there are not explanations among them. There are not reasons why T2 decrease pH and increase Lactic acid.
Author Response
All the suggested changes given in pdf were incorporated.
Round 2
Reviewer 1 Report
Accept in present form.
Reviewer 2 Report
The new version of the manuscript shows that the authors has considered the suggested modifications, therefore is suitable for publication.